# *Arabidopsis thaliana* Accessions from the Chernobyl Exclusion Zone Show Decreased Sensitivity to Additional Acute Irradiation

**DOI:** 10.3390/plants11223142

**Published:** 2022-11-17

**Authors:** Mikhail Podlutskii, Darya Babina, Marina Podobed, Ekaterina Bondarenko, Sofia Bitarishvili, Yana Blinova, Ekaterina Shesterikova, Alexander Prazyan, Larisa Turchin, Dmitrii Garbaruk, Maxim Kudin, Gustavo T. Duarte, Polina Volkova

**Affiliations:** 1Russian Institute of Radiology and Agroecology, 249032 Obninsk, Russia; mikhail.podlutskii@gmail.com (M.P.); babinadd@gmail.com (D.B.); podobedmyu@gmail.com (M.P.); bondarenco.e@gmail.com (E.B.); bitarishvili.s@gmail.com (S.B.); yana.manuhina@yandex.ru (Y.B.); eshesterikova89@gmail.com (E.S.); prazyan22@gmail.com (A.P.); 2Polesye State Radiation-Ecological Reserve, 247618 Khoiniki, Belarus; turchin2006@bk.ru (L.T.); dima.garbaruk.77@mail.ru (D.G.); max.kudin@mail.ru (M.K.); 3Belgian Nuclear Research Centre (SCK CEN), Unit for Biosphere Impact Studies, 2400 Mol, Belgium; gtduarte@sckcen.be; 4Independent Researcher, 2440 Geel, Belgium

**Keywords:** RNA sequencing, chlorophyll fluorescence, radioadaptation, radioecology, low-dose radiation, chronic irradiation, plant radiobiology

## Abstract

Chronic ionising radiation exposure is a main consequence of radioactive pollution of the environment. The development of functional genomics approaches coupled with morphological and physiological studies allows new insights into plant adaptation to life under chronic irradiation. Using morphological, reproductive, physiological, and transcriptomic experiments, we evaluated the way in which *Arabidopsis thaliana* natural accessions from the Chernobyl exclusion zone recover from chronic low-dose and acute high-dose γ-irradiation of seeds. Plants from radioactively contaminated areas were characterized by lower germination efficiency, suppressed growth, decreased chlorophyll fluorescence, and phytohormonal changes. The transcriptomes of plants chronically exposed to low-dose radiation indicated the repression of mobile genetic elements and deregulation of genes related to abiotic stress tolerance. Furthermore, these chronically irradiated natural accessions showed higher tolerance to acute 150 Gy γ-irradiation of seeds, according to transcriptome and phytohormonal profiles. Overall, the lower sensitivity of the accessions from radioactively contaminated areas to acute high-dose irradiation may come at the cost of their growth performance under normal conditions.

## 1. Introduction

Ionising radiation (IR) is an abiotic stress factor whose relevance in the modern biosphere is determined mainly by anthropogenic activity, including nuclear tests, disasters at nuclear power plants (NPPs), mining, and radioactive waste. The biological effects of chronic radiation exposure are of utmost interest [1,2,3] since chronic long-term irradiation is the main consequence of radioactive pollution of the environment. Areas contaminated after radiation disasters represent unique sites for studying the influence of heterogenic chronic irradiation on plant populations in their natural environment.

IR is a genotoxic factor that can directly damage DNA and other biomolecules, or trigger indirect damage through radiolysis of cellular water and extensive production of reactive oxygen species (ROS). Several recent reviews describe in detail the effects of IR on plants [4,5,6]. Constant genotoxic exposure and increased levels of mutagenesis may speed up the microevolution of plant populations growing in radioactively contaminated areas. The adaptation to IR of chronically irradiated species can be tested in laboratory-controlled conditions using either acute or chronic radiation exposure. The development of “omics” approaches and the extension of genomic databases alleviate application of modern molecular tools to the radioecology field [7]. The use of model species remains a standard for genomic research and can provide deeper insights into the molecular changes under adverse environmental conditions. 

One of the main sites for investigating chronic radiation exposure is the Chernobyl exclusion zone (CEZ), where populations of plants and animals have been studied since the explosion at the nuclear power plant in 1986. Some of that research includes populations of the model plant *Arabidopsis thaliana* L. which is considered as a radioresistant species (LD_50_ exceeds 600 Gy) [8]. *A. thaliana* natural accessions inhabiting a 30 km radius of the Chernobyl NPP were analysed for the frequency of embryonic lethal mutations in 1987 and 1988, and the plots that retained high levels of radioactive contamination long after the initial exposure had a higher frequency of mutant plants [9]. Progeny of plants collected from 1986 to 1992 resisted higher concentrations of mutagens having more than 10-fold lower frequency of extrachromosomal homologous recombination and a higher level of global genome methylation [10]. However, a later work showed a decrease in the level of genome-wide DNA methylation following an increasing dose rate for *A. thaliana* plants sampled in Chernobyl in 2016 [11]. However, chronic radiation exposure did not influence growth parameters or the cellular antioxidant status of shoots in the progeny of Chernobyl *A. thaliana* plants grown under normal conditions [12]. Interestingly, another Chernobyl natural accession was more tolerant to cadmium treatment, as reflected by proteomic alterations of energy production and ascorbate–glutathione cycle [13].

However, the information regarding functional changes in *A. thaliana* accessions growing under chronic irradiation for decades is still incomplete. Plant genomic data from the Chernobyl zone is still scarce, and the microevolution processes in this radioactively contaminated area remain to be explored. It is still unclear if life in these heterogeneous chronic radiation exposure conditions may lead to an increase in radioresistance of the organisms. In this work, we applied transcriptomic, reproductive, and physiological approaches for evaluating the way in which *A. thaliana* natural accessions from CEZ recover from chronic low-dose and acute high-dose γ-irradiation of seeds and if these accessions can be considered better adapted to additional radiation exposure.

## 2. Results

### 2.1. Arabidopsis Natural Accessions in the Chernobyl Exclusion Zone

Seeds of *A. thaliana* plants were collected at three experimental plots (Figure 1) in the Polesye State Radiation-Ecological Reserve (Khoiniki, Gomel Region, Republic of Belarus) in June 2019. The sampling details are provided in Appendix A. For each plot, between 10 and 20 whole dry shoots with ripened seeds were combined as a single accession. Soils of experimental plots were podzolic with slopes in a range of 0–15%, and the levels of available potassium (Table 1) ranged from low (Babchin) to insufficient (Vygrebnaya Sloboda and Masany) [14], which may facilitate the accumulation of caesium in tissues of plants from radioactively contaminated plots [15]. The experimental plots had rather different levels of radioactive contamination (Table 2).

Babchin plot (accession Bab-0) was used as a reference because of almost background radiation levels. The Vygrebnaya Sloboda plot (accession VS-0) had relatively low levels of contamination, and the Masany plot (accession Masa-0) was the most contaminated (Table 2). Part of the collected seeds was further γ-irradiated at a dose of 150 Gy (dose rate 460 Gy × h^−1^). All seeds were germinated and grown on half-strength MS media under controlled conditions. Hereinafter, these natural accessions will be referred to as Bab-0, VS-0, and Masa-0, respectively, and all assays involving seeds or seedlings will be referred to as reference, recovery from chronic low-dose (LD), or recovery from acute high-dose (HD) radiation exposure. Germination rates, chlorophyll fluorescence, and transcriptional profiles of seedlings of additionally irradiated (acute exposure) and non-irradiated (chronic exposure) seeds were analysed.

### 2.2. Germination Dynamics Analysis

Germination dynamics was assessed by analysing time, rate, homogeneity, and synchrony parameters. Germination percentage (expressed as the percent ratio of the number of germinated seeds to the total seeds sown) showed significant differences between the chronically irradiated accessions VS-0 and Masa-0 and the reference accession Bab-0 (Figure 2A). Seeds from radioactively contaminated plots showed significantly lower germinability (−41% for Masa-0 and −34% for VS-0) than seeds from the Babchin reference plot. Germinability assessed by days revealed that the difference between the natural accessions become evident on the 3rd day. Acute exposure to 150 Gy increased Bab-0 seeds germination (+17%), which became significantly different on the 5th day (Figure 2B). Interestingly, the acute γ-irradiation of seeds from contaminated CEZ areas had no impact on germinability, although in general it did not exceed 40% after 6 days for both acutely and chronically irradiated seeds (Figure 2B).

Germination Rate Index [16], which represents the total number of germinated seeds at the end of the test divided by the mean germination time, was significantly lower for seeds of accessions VS-0 (−43%) and Masa-0 (−48%) compared to Bab-0 (Figure 2C). Altogether, we observed lower responsiveness to acute high-dose IR of both accessions coming from chronically irradiated areas (VS-0 and Masa-0), while a small hormetic effect was evidenced on Bab-0 seeds.

### 2.3. Leaf Area Analysis

Next, we evaluated the leaf area of 11-day-old seedlings of the three Chernobyl accessions. Chronically exposed Masa-0 and VS-0 accessions showed a significantly smaller leaf area (−39% and −55%, respectively) than Bab-0 (Figure 3). However, after acute high-dose γ-radiation exposure, the mean leaf area of Bab-0 seedlings was significantly decreased (−46%, compared to non-irradiated Bab-0), while no significant change was detected for both VS-0 and Masa-0 (Figure 3). These results indicate that the small hormetic effect on Bab-0 seed germination after acute IR exposure is in detriment of leaf area, while both Masa-0 and VS-0 accessions show a significantly lower IR sensitivity.

### 2.4. Photosynthetic Parameters

Fv/Fm ratio provides an estimate of the maximum quantum efficiency of PSII photochemistry. In plants recovering from chronic IR exposure, the median values of this parameter were significantly lower than those of the reference Bab-0 accession, which can be associated with stress response and a decrease in photosynthetic intensity [17]. VS-0 plants showed significantly lower values of Fm′ and higher of qL (coefficient of photochemical fluorescence quenching) compared to Bab-0 plants (Figure 4), which together with a decrease in Fv/Fm suggested a more oxidized state of Q_A_ (i.e., higher fraction of open centres) [18], accompanied with a higher efficiency of non-photochemical quenching [19], pointing to the lower photosynthetic efficiency in this accession.

Photosynthetic parameters reflect the overall physiological condition of plants in the extreme environment [20], and chronic radiation exposure can influence them in a species-specific way [3]. Significant photosynthetic changes after acute high-dose γ-irradiation of seeds were found only for Bab-0 plants (Figure 5, measurements for other accessions are provided in Appendix A). Maximum fluorescence levels Fm and Fm′ in the dark-acclimated and light-exposed samples, respectively, as well as the minimal fluorescence yield in light-exposed leaves F0′ were significantly higher in plants developed from acutely irradiated Bab-0 seeds (Figure 5), probably pointing to slight stimulation of photosynthesis after acute irradiation.

### 2.5. Phytohormone Measurements

Since phytohormones mediate a wide range of adaptive responses to stressors, including chronic irradiation, abscisic acid (ABA), zeatin (cytokinin), salicylic acid, indole-3-acetic acid (IAA), and indole-3-butyric acid (IBA), concentrations and their ratios were assessed using high-performance liquid chromatography (HPLC; Figure 6 and Figure 7).

ABA is considered as the main phytohormone controlling signaling pathways involved in the response to biotic and abiotic stress factors. The HPLC analysis revealed that ABA concentrations in *A. thaliana* plants increased after acute γ-irradiation in all the samples (Figure 6A and Figure 7A).

Auxins and cytokinins belong to growth-promoting hormones. Zeatin, a representative of the cytokinin class, regulates the cell cycle, cell division, and the development of the shoot apical meristem. Auxins control plant tropisms, extensional growth, and development of the root meristem [21]. The concentrations of zeatin and IAA were higher in VS-0 and Masa-0 plants recovering from chronic IR in comparison to Bab-0 (Figure 6). While IAA increased +346% for Masa-0, and +68% for VS-0 (Figure 6B), zeatin increased +91% and +51%, respectively (Figure 6E). Conversely, zeatin and IAA concentrations reduced in Masa-0 and VS-0 plants recovering from acute high-dose γ-irradiation in comparison to their control counterparts. IAA decreased −86% for Masa-0, and −81% for VS-0 (Figure 6B and Figure 7A), while zeatin decreased −67% and −29%, respectively (Figure 6E and Figure 7A). These concentrations almost did not change between acute-irradiated and control Bab-0 plants. Unlike IAA, the concentration of IBA in Bab-0 plants was threefold the value of IBA in samples after γ-irradiation (0.12 and 0.04 µM, respectively) (Figure 6C).

Salicylic acid is a key phytohormone involved in the plant immunity. In our study, the maximum salicylic acid concentrations were observed in Bab-0 plants from the background conditions, and were much lower in samples from radioactively contaminated plots (−87% for Masa-0 and −95% for VS-0, compared to Bab-0) (Figure 6D). 

Verma et al. [22] highlighted the significance of crosstalk between different hormones in generating a sophisticated and efficient stress response. Therefore, we evaluated the ratios of IAA/IBA, salicylic acid/IAA, zeatin/salicylic acid, IAA/zeatin and ABA/zeatin as a result of acute γ-irradiation (Figure 7B). All the analysed ratios showed a clear differentiation between the radioactively contaminated plots and the plot with normal radioactive background. The only exception was the ABA/zeatin ratio, which significantly increased as a result of γ-irradiation in samples of all the studied plots, reflecting ABA accumulation and growth retardation after irradiation.

### 2.6. Differential Gene Expression Analysis of Control-Grown Chernobyl Natural Accessions

Because both VS-0 and Masa-0 accessions have been experiencing chronic radiation exposure for decades, we compared their transcriptome profiles to the reference Bab-0 (Figure 8). In the accession from the less contaminated plot (VS-0, Figure 8A), 452 differentially expressed genes (DEGs) were identified, of which 186 were upregulated and 266 were downregulated (log_2_FC ≥ |2|). The highest log_2_FCs were registered for *AT3G42723* (ATP binding protein), *AT4G36140* (disease resistance protein), *AT3G44070* (glycosyl hydrolase family 35 protein), *AT2G29300* (NAD(P)-binding Rossmann-fold superfamily protein), and *AT5G66052* (transmembrane protein) (Appendix A). The lowest log_2_FCs were found for *AT3G61030* (calcium-dependent lipid-binding family protein), *AT5G17890* (DA1-related 4 protein), *AT2G41440* (agamous-like MADS-box protein), *AT5G17880* (disease resistance-like protein), and *AT1G23915* (hypothetical protein) (Appendix A). Next, Gene Ontology (GO) enrichment analysis was performed in order to understand if the observed responses were related to common regulatory pathways (Table 3). Functional enrichment was only found among downregulated genes for VS-0 accessions recovering from chronic irradiation (Table 3 and Appendix A). Significant GO enrichment was found mainly for the protein phosphorylation processes and responses to abiotic and biotic stressors. Enrichment in Molecular Function categories included protein serine/threonine kinase activity, ADP and ATP binding, calcium ion binding.

In comparison to the reference Bab-0, 222 DEGs were identified in *A. thaliana* seedlings recovering from the chronic irradiation at the most contaminated plot Masany (Masa-0, Figure 8B). Among them, 156 genes were upregulated, and 66 genes were downregulated. The highest levels of differential expression were found for *AT1G14250* (probable apyrase 5), *AT5G35480* (hypothetical protein), *AT2G29300* (NAD(P)-binding Rossmann-fold superfamily protein), *AT4G19500* (disease resistance protein), and *AT2G27402* (plastid transcriptionally active protein) (Appendix A). Among the 66 downregulated genes, the strongest responses were observed for the genes *AT1G25083* (glutamine amidotransferase type 1 family protein), *AT1G23915* (hypothetical protein), *AT2G41440* (agamous-like MADS-box protein), *AT1G15640* (transmembrane protein), and *AT3G53840* (wall-associated receptor kinase-like 15) (Appendix A). Enriched GO terms in Masa-0 seedlings recovering from chronic IR exposure were related to ADP binding (upregulated genes), protein serine/threonine kinase activity, and carbohydrate binding (downregulated genes) (Table 3 and Appendix A).

In order to understand if common response patterns could be identified between the accessions from chronically irradiated plots, we compared overlapping responses between the DEGs. In total, VS-0 and Masa-0 lines had 105 shared DEGs, of which 88 were upregulated and 32 downregulated (Appendix A). According to the GO annotation, the commonly regulated genes were mostly related to phytohormonal responses, transcriptional and translational control (including long non-coding RNAs (lncRNA)), defence responses, oxidation–reduction processes, signal transduction, and plant development. However, there were no enriched GO terms among the control-grown VS-0 and Masa-0 overlapping genes, which may suggest that they are under unique developmental programs, although using a common set of regulators.

### 2.7. Differential Gene Expression Analysis of Chernobyl A. thaliana Accessions Recovering from Acute High-Dose γ-Irradiation

Next, we asked whether the accessions chronically exposed to IR in the CEZ would show different transcriptome profiles in response to acute high-dose γ-irradiation. Therefore, we subjected Bab-0, VS-0 and Masa-0 seeds to 150 Gy γ-irradiation, and analysed the transcriptional profiles of 13-day-old recovering seedlings in comparison to their respective non-irradiated counterparts (Figure 9).

Recovering plants from the non-contaminated Bab-0 accession had 276 upregulated and 202 downregulated DEGs after γ-irradiation of seeds (Figure 9A, Appendix A). Among the upregulated ones, the strongest responses were observed for *AT3G12230* (serine carboxypeptidase-like 14), *AT1G65483* (hypothetical protein), *AT2G28680* (RmlC-like cupins superfamily protein), and *AT1G28650* (GDSL-like lipase/acylhydrolase superfamily protein). The most downregulated genes were *AT1G33280* (NAC domain containing protein 15), *AT1G05660* (pectin lyase-like superfamily protein), *AT4G19770* (glycosyl hydrolase family protein with chitinase insertion domain), and *AT3G58060* (cation efflux family protein). No DNA repair-related genes were identified among the DEGs. Overall, enriched GO terms were related to stress (including oxidative damage), hormone response, and developmental processes, besides involving cellular components such as cell wall and membranes (Appendix A), both among up- and downregulated genes.

Interestingly, the highest number of DEGs after γ-irradiation of seeds were identified for the VS-0 recovering seedlings from the low contaminated CEZ plot, with 671 genes in total (Figure 9B). Of those, 360 were upregulated and 311 were downregulated (Appendix A). The most upregulated genes included *AT3G60120* (beta glucosidase 27), *AT1G35320* (unknown protein), *AT1G65483* (unknown protein), *AT3G25180* (cytochrome P450), and the most downregulated ones included *AT1G52820* (2-oxoglutarate and Fe(II)-dependent oxygenase superfamily protein), *AT5G06900* (cytochrome P450), *AT1G15540* (2-oxoglutarate and Fe(II)-dependent oxygenase superfamily protein), *AT4G11310* (papain family cysteine protease). Two DNA helicases involved in repair processes were identified: *AT1G09995*, induced by the treatment, and *AT5G43530*, repressed. The enriched GO terms for VS-0 and Bab-0 seedlings recovering from acute high-dose IR exposure were similar, both among up- and downregulated genes (Appendix A), comprising elements involved in stress response, hormone signaling, and cellular components such as membranes and cell wall.

Significantly fewer DEGs were identified in seedlings of the accession Masa-0 from the heavily contaminated plot during their recovery from the acute γ-irradiation of seeds (Figure 9C), with 33 upregulated and 13 downregulated (Appendix A) genes. Among the most upregulated genes were *AT5G49420* (MADS-box transcription factor family protein), *AT1G15640* (unknown protein), *AT5G14160* (F-box family protein), and *AT4G37990* (elicitor-activated gene 3-2). The strongest repression levels were observed for *AT2G27402* (unknown protein), *AT2G29000* (leucine-rich repeat protein kinase family protein), *AT3G61030* (calcium-dependent lipid-binding family protein), *AT3G62460* (putative endonuclease or glycosyl hydrolase). All of Masa-0 enriched GO terms were among the upregulated genes (Appendix A); however, as for Bab-0 and VS-0, they comprised stress and hormone response pathways, along with cell wall component.

Next, we searched for common response patterns on seedlings recovering from acute γ-irradiation of seeds across the three Chernobyl natural accessions in order to identify unique and common profiles among the IR-sensitized (VS-0 and Masa-0) and non-sensitized (Bab-0) accessions (Figure 10). Twenty DEGs were shared among all experimental plots (Appendix A), nineteen of which were induced and one was repressed. Among them were genes related to antioxidant defence and oxidation–reduction processes (*GSTU3, GSTU12, MSRB7, AT1G13340, SRG1, GSTF7, NEET*), transcription factors (*AT5G49420, ATMYB2, WRKY75*), stress response (*PYD4, AT1G74010*), mucilage biosynthetic process (*AT3G10320*), and defence response (*CAD8, PMEI10, SBT3.3, XTH25, KTI1, PR1, ACA12*).

Seedlings of the accessions Bab-0 and VS-0 had similar transcriptional profiles after acute γ-irradiation of seeds (Table 4 and Appendix A). Common GO terms include responses to ROS, salicylic acid, jasmonic acid, ABA, peroxidase activity and heme binding, defence responses to bacteria and fungi, responses to wounding, salt stress, hypoxia, metabolism of hydrogen peroxide and metabolism of cell wall components, including xyloglucans, suberin, and lignin.

Compared to the other plots, few enriched GO terms were revealed for the seedlings of Masa-0 accession (Table 4 and Appendix A). Those included immunity and phytohormone-related responses. All additionally irradiated plants showed enrichment in terms associated with cell wall (Table 4).

The analysis of overlapping DEGs (Figure 10) revealed no GO terms were enriched among the overlapping DEGs of three accessions recovering from acute γ-irradiation of seeds. However, GO enriched terms were found among the overlap between Bab-0 and VS-0 (Figure 10). Among upregulated genes, these two accessions shared several enriched terms including “Response to oxidative stress, “Response to jasmonic acid”, “Jasmonic acid hydrolase”, “Xyloglucan:xyloglucosyl transferase activity”. Enriched GO terms for downregulated overlapping genes included “Hydrogen peroxide catabolic process”, “Root morphogenesis”, “Stem cell population maintenance”, “Cell wall”. The complete list of enriched GO terms is provided in Appendix A.

Overall, these results suggest that both Bab-0 and VS-0 seedlings adopt similar strategies for coping with the acute high-dose irradiation of seeds. Furthermore, it is possible that Masa-0 accession, which derives from plants that were exposed to a higher degree of chronic IR at the CEZ, tolerates higher levels of seeds γ-irradiation, which could reflect an adaptation process. Interestingly, among the 21 Masa-0-specific DEGs, 10 were also found during the recovery from chronic radiation exposure, but with an opposite expression pattern (Table 5).

## 3. Discussion

*A. thaliana* is an annual plant, which means that at least 35 generations have passed since Chernobyl NPP explosion in 1986. Although the radiation levels continuously decrease due to the radioactive decay, the development under chronic IR exposure supposedly left traces in the genome of studied species, either at genetic or epigenetic levels. Since chronic radiation exposure is a genotoxic factor which is also able to provoke oxidative stress at the cellular level, one might expect changes over time in the fine tuning of DNA repair and ROS balancing. Antioxidant responses were indeed identified in our earlier studies [3,23]. However, we also discovered strong transcriptional activation of plant innate immune responses after irradiation, which can be associated with the cell wall damage by gamma quanta and electrons during radiation exposure. Analysis of reproductive, physiological, and transcriptional responses of the model plant *A. thaliana* can reveal new adaptive reactions of plants to chronic IR, since *A. thaliana* genome is annotated significantly better than genomes of other plant species.

### 3.1. Growth and Physiological Responses of Chernobyl Accessions to Chronic and Acute Irradiation

Under control conditions, the natural accessions from radioactively contaminated areas, VS-0 and Masa-0, had suppression in germination (Figure 2) and growth (Figure 3) efficiencies when compared to the reference accession, Bab-0. The quantum efficiency of open photosystem II centres (Fv/Fm) was also significantly lower in VS-0 and Masa-0 (Figure 4 and Figure 5), suggesting the repression of photosynthetic capability, which may be reflected as growth retardation. The changes in phytohormonal balance (Figure 6 and Figure 7) are also of outmost importance for plant growth performance under stress conditions [24,25]. A survey including 30 different *A. thaliana* ecotypes showed that plant stress response was often associated with an increase in the ABA/cytokinin ratio [26]. We also observed an increase of ABA/zeatin ratio for chronically irradiated accessions (Figure 7B), which may point to stress responses. The same authors showed that cytokinin levels can be used as an ecotype classifier [15], and in our study, zeatin concentrations in Masa-0 and VS-0 plants recovering from chronic irradiation were higher than those in the reference Bab-0 accession (Figure 6), indeed suggesting that Chernobyl *A. thaliana* accessions are substantially different from each other. 

In order to understand if microevolutionary processes in the Chernobyl exclusion zone led to an increase in tolerance to radiation exposure, we applied acute γ-radiation (150 Gy) to seeds of Chernobyl natural accessions and evaluated seedling recovery using germination, photosynthesis, and transcriptomic analyses. In comparison to untreated seeds, a hormetic effect after acute γ-irradiation of Bab-0 reference seeds was observed by a better germination rate (Figure 2A), but at the cost of seedling leaf area (Figure 3). These plants also showed slightly higher photosynthetic activity (Figure 4). Conversely, no significant changes were observed for either Masa-0 or VS-0 after acute γ-irradiation, despite their naturally poor germinability and smaller leaf area (Figure 2 and Figure 3), which may be considered as a sign of radioadaptation. Indeed, in natural plant populations inhabiting polluted areas, adaptation to adverse conditions can be associated with deteriorated performance in normal conditions, while their increased fitness becomes evident under stress exposure [27]. Significant photosynthetic changes after acute high-dose γ-irradiation of seeds were found only for Bab-0 plants (Figure 5, Appendix A). It has been hypothesized that the modulation of photosynthetic parameters may overlap with IR responses due to the necessity of controlling the levels of ROS [3,28]. While ROS are naturally produced during cell metabolism and especially photosynthesis, under IR exposure their levels increase. This finding also suggests a higher degree of IR tolerance of both VS-0. and Masa-0 accessions, as they were photosynthetically unresponsive to acute high-dose irradiation.

Phytohormonal changes were observed in all additionally irradiated accessions. The ratio IAA/IBA decreased for both chronically irradiated accessions after additional acute irradiation (Figure 7B) accompanied with a decrease in auxin concentrations, especially IAA (Figure 6). Decreased levels of active IAA in certain stress conditions can increase plant stress tolerance through growth inhibition [29]. The decrease in IBA content as an IAA precursor may be triggered by stress-induced ROS overproduction, which promotes conjugation of IBA and glucose, subsequently decreasing the levels of IAA [30,31]. Increased IAA concentrations in Bab-0 plants may therefore reflect more limited ability of this accession for adaptation to high-dose radiation exposure.

In addition, in Bab-0 plants recovering from acute γ-irradiation, zeatin concentrations did not change while they considerably decreased in Masa-0 and VS-0. The ABA/zeatin ratio, on the other hand, increased after acute γ-irradiation in all the samples, and to a higher degree in Masa-0 plants. An increase in the ABA to CKs ratio is linked to stomatal aperture size regulation, and it is frequently associated with plant response to stress factors [26,32]. Cytokinins crosstalk with ABA functions in drought and salinity stress responses [22,33]. There is evidence that mutual regulation mechanisms exist between the cytokinins and ABA metabolism, and signals regulating plant growth and development [33]. All cytokinin-deficient plants with reduced levels of various cytokinins exhibited a strong abiotic stress-tolerant phenotype that was associated with increased cell membrane integrity and ABA hypersensitivity [33]. Therefore, the decrease in zeatin concentrations in natural accessions VS-0 and Masa-0 and increased ABA/CKs ration may reflect an ability to better tolerate additional radiation exposure.

### 3.2. Transcriptional Responses of Natural Accessions Recovering from Chronic Radiation Exposure

Significant transcriptomic differences among the natural accessions recovering from chronic IR exposure (Masa-0 and VS-0) were scored in comparison to the reference Bab-0. While the GO enrichment analysis suggested that VS-0-recovering plants showed a stress-response profile, which included responses to ROS, such enrichment was not evident in Masa-0 seedlings. The 105 overlapping genes between VS-0 and Masa-0 (Appendix A) which most likely reflect adaptive responses to chronic irradiation exposure were also not significantly enriched in specific pathways. This observation suggests that these accessions may be under different developmental programs, and that recovery from chronic low-dose irradiation may not rely on specific pathways, but rather on distinct regulatory elements. Such a pattern has been noticed also for Scots pine and *Capsella bursa-pastoris* chronically exposed to IR [3,34]. Nevertheless, among the overlapping genes were those related to phytohormonal responses, gene expression control, immune responses, antioxidant system, and plant development. Among the strongest responses observed, upregulated *AT2G29300* encodes putative tropinone reductase, and overexpression of this gene was observed in the leaves of *A. thaliana* subjected to combination of drought and heat stress [35]. Its orthologue in *Brassica napus*, *BnTR1*, is related to transpiration, while overexpressing plants show increased transpiration rate and enhanced low temperature tolerance under freezing conditions [36]. Downregulated *AT1G23915* encodes an unknown protein, the sequence of which resembling, according to the Protein BLAST, the MULE transposase domain. 

Specific groups of genes of interest can be identified in the set of commonly regulated DEGs between VS-0 and Masa-0, such as defence responses, deregulation of cytochromes, glycosyl hydrolases, set of F-box proteins, regulators of programmed cell death, and various long non-coding RNAs. In order to exclude the influence of environmental factors, we ran an analysis of this set using Genevestigator platform (https://genevestigator.com/, accessed on 3 November 2022) that revealed similarities with experiments related to drought responses, and, specifically, to caesium (data deposited under accession number PRJNA99145) and γ-irradiation [37]. The lower potassium levels in radioactively contaminated soils (Table 1) can, in fact, contribute to a higher caesium accumulation in the plants as a result of reduced competition during root uptake [15]. 

Overall, these findings confirm earlier works where the modulation of mobile genetic elements and induction of genes related to abiotic stress tolerance were found under chronic radiation exposure [3,34].

### 3.3. Transcriptional Responses of Natural Accessions to Acute Radiation Exposure

In order to identify common regulators that could play a role on the recovery from acute γ-irradiation, and for evaluating if the radioactive contamination history of natural accessions would impact on the response levels, we first evaluated the overlapping genes among 150 Gy-irradiated Bab-0, Masa-0, and VS-0 plants having as reference their respective non-acutely irradiated counterparts. Interestingly, the overlap between the three accessions was low, with only 20 DEGs related to antioxidant defence, general stress response pathways, and cell wall responses. They may represent markers for recovery from acute γ-irradiation, but also suggest that the control of oxidative stress damage may be a long-lasting process and that persistently elevated levels of ROS can occur long after radiation exposure [38].

The low overlap between the three accessions was mainly due to accession Masa-0 that had fewer DEGs and a different expression profile (Figure 8, Table 4). This observation may indicate that this natural accession has become somewhat insensitive to high γ-radiation levels, as also suggested by phenotypic parameters (Figure 2B and Figure 3). The 21 Masa-0-specific DEGs may also reflect essential regulatory elements for a stress-primed organism. Interestingly, among them, 10 were also deregulated on Masa-0 plants recovering from chronic radiation exposure when compared to the background reference Bab-0, but with an opposite expression pattern (Table 5). These data confirmed our previous findings [34] that acute and chronic irradiations often trigger opposite transcriptional responses patterns. Among these 21 unique DEGs, especially relevant seems upregulation of stress-responsive bHLH transcription factor *AT5G39860*, which has a regulatory role in gibberellin-dependent development [39], rapidly responds to shade [40] and to hydrogen peroxide [41]. Downregulated *AT3G30775* encodes for proline oxidase ProDH1, which is repressed by osmotic stress [42]. Deficiency in this enzyme leads to prolonged accumulation of proline during stress [43], while free proline takes part in antioxidant defence and stress signaling [44]. The calcium/lipid-binding endonuclease/exonuclease/phosphatase *AT3G60950* is known to interact with Tdp1 proteins, which are involved in DNA repair [45]. Other Masa-0-specific genes deregulated during the recovery from the acute γ-irradiation may also reflect the hormonal imbalance in these plants, such as *AT1G15640*, which encodes auxin and cytokinin cross-talk component [46], and SAUR-like auxin-responsive proteins *AT5G18010*, *AT5G18050*, *AT5G18080*, *AT5G18030* that play a central role in auxin-induced acid growth [47,48].

On the contrary, seedlings of background reference Bab-0 and VS-0 accession from the plot with low level of contamination had similar transcriptional responses to acute high-dose irradiation. Their transcriptional profiles had 248 commonly regulated DEGs (Appendix A). Such an overlap was also reflected by the enrichment of GO terms, which included responses to ROS, salicylic acid, jasmonic acid, ABA, peroxidase activity, defence responses, and metabolism of cell wall components (Table 4 and Appendix A). The higher overlap between Bab-0 and VS-0 transcriptional profiles to acute γ-irradiation also indicates that differences in soil composition are not the main driver of the response patterns observed in this work, since Masa-0 and VS-0 were characterized by more similar soil conditions (Table 1). 

All plants recovering from acute γ-irradiation showed enrichment in terms associated with cell wall (Table 4). Cell wall modifications were evident for several transcriptomic profiles of other plants after irradiation [3,23,34], suggesting that penetration of the cell wall by γ-quantum and particles can induce responses similar to wounding of pathogen attack.

## 4. Conclusions

Studies of microevolutionary consequences of chronic radiation exposure for plant populations remain important in the light of development of new nuclear technologies, radionuclide remediation strategies, and space research. In this work, we observed that *A. thaliana* natural accessions from the Chernobyl exclusion zone had worse growth performance than the background accession. The growth differences were accompanied by changes in phytohormonal balance and chlorophyll fluorescence parameters. However, accessions from the radioactively contaminated plots demonstrated lower sensitivity to acute high-dose γ-irradiation of seeds, which may reflect an adaptive process after generations of chronic radiation exposure. Transcriptional profiles indicated a clear difference between the expression pattern of plants recovering from chronic or acute radiation exposure. The accession Masa-0 from the plot with high level of heterogenous radioactive contamination had few differentially expressed genes after acute γ-irradiation of seeds, showing more specific response to γ-irradiation and being probably adapted to higher doses of exposure. Ongoing study of specific single nucleotide polymorphisms may help to uncover the source of such adaptation.

Transcriptional responses of *A. thaliana* to chronic and acute irradiation are in good agreement with transcriptional studies of other plant species, suggesting opposing response patterns to acute and chronic irradiation, and important roles of antioxidant system, chaperones, cell wall responses, and mechanisms of DNA protection in withstanding ionising radiation stress. Since ionising radiation triggers continuous primary and secondary ROS production, damages DNA, membranes, and proteins, the activation of abovementioned defence systems can ensure the success of plant populations in radioactively contaminated areas and provide the direction of candidate gene search for space missions, where high radiation load is expected.

## 5. Materials and Methods

### 5.1. Sampling in the Chernobyl Exclusion Zone

Seeds of naturally growing *A. thaliana* plants were collected at three experimental plots on the territory of Polesye State Radiation-Ecological Reserve (Khoiniki, Gomel Region, Republic of Belarus) with different radioactive contamination levels in June 2019 (Figure 1). Each accession was composed by a pool of 10–20 plants harvested in each plot. The Babchin plot (accession Bab-0) was used as reference because of the background level of radioactivity. Vygrebnaya Sloboda (accession VS-0) was considered as low-contaminated plot, and Masany (accession Masa-0) as the most contaminated plot. Photos of experimental plots are provided in Appendix A. The weather conditions during sampling varied as following: ambient temperature—from 29.5 °C to 34.0 °C, relative humidity—from 48% to 50%. There was no precipitation during sampling and a week before in the area. The experimental plots were similar (open areas, uniform insolation).

The radiological parameters of the experimental plots are given in Table 1. The α- and β-particles flux densities (min^−1^ × cm^−2^) were determined at the levels of 0.2 cm from the ground as well as ambient dose rate measurement (μSv × h^−1^) were determined on a distance of 1 m from the surface and any surrounding subjects using a dosimeter-radiometer MKS-02SA1 (SNIIP, Moscow, Russian Federation). The relative error of dose rate and flux densities measurements did not exceed 20%. 

At each plot, we collected soil samples for the assessment of physical and chemical properties, radionuclide content, and heavy metal contamination. The soil was collected by “envelope” method, taking 15 cm of samples from the corners and the centre of a rectangular area (1 × 2 m) inside the plot and mixing them into a pooled sample. In the soil samples, in accordance with the ISO standard for soil quality (ISO/TC 190) we analysed pH, hydrolytic activity, K, P, and Ca contents, cation exchange capacity, and humus content (Appendix A).

Total concentrations of heavy metals (Cd, Cu, Co, Ni, Cr, Mn, Pb, Zn, As, Mo) (Appendix A) were measured in the soil samples using plasma optical emission spectrometer (ICP-OES, Varian, Mulgrave, Australia) in accordance with the ISO 11,047 standard, as previously described [34]. 

The active concentrations of ^137^Cs were measured with a γ-spectrometer CANBERRA (Atlanta, Georgia, USA) with a coaxial semiconductor Ge (Li) detector and an extended energy range. To determine ^90^Sr activity in soil, radiochemical method was used.

### 5.2. Acute γ-Irradiation of Seeds

To identify a presumable radioadaptation of *A. thaliana* populations from plots with different levels of radioactive contamination, the seeds were exposed to acute γ-radiation at a dose of 150 Gy [49] using scientific irradiation facility «GUR-120» (Russian Institute of Radiology and Agroecology, Obninsk, Russian Federation), ^60^Co, with a dose rate of 460 Gy × h^−1^. For phytohormonal, morphological, and reproductive assessments, two independent experiments were carried out with four biological replicates per plot (20 seeds per replicate) in each.

### 5.3. Growth Conditions and Germination Assay

The seeds were planted for germination after stratification at 4 °C for 7 days. Half-strength solid (0.5% agar) Murashige–Skoog nutrient medium (supplemented with 0.3% sucrose) was used to grow seedlings of *A. thaliana*. Seed germination was carried out in a plant growth chamber SANYO MLR-351H (SANYO Electric Co., Ltd., Osaka, Japan) under the following conditions: long daylight hours (16 h light/8 h darkness), temperature of +21 °C, relative humidity—55%, the density of the photosynthetic photon flux—80 μmol photons m^−2^ × s^−1^. 

Seed germination was assessed as endosperm rupture and visible radicle emergence during the first 6 days after transfer to the plant growth chamber. 

Germination parameters were calculated using the Germinationmetrics package v0.1.3 for RStudio v1.4 using data from a partial germination count. For statistical analysis of experimental data and visualization of the results, tidyverse and rstatix packages for R were used. A nonparametric analysis of data variance was carried out using the Kruskal–Wallis test to compare the studied groups. For significant results, an additional Dunn’s test was performed (using the Holm–Bonferroni correction for multiplicity).

### 5.4. Leaf Area and Fluorescence Measurements

The photosynthetic activity was assessed for 17-day-old juvenile plants grown in conditions described in Section 5.3. All fluorescence measurements were performed using a pulse–amplitude modulated chlorophyll (Chl) fluorometer Junior-PAM (Heinz Walz GmbH, Effeltrich, Germany). The fluorescence parameters were measured for 54 samples (three accessions (Bab-0, VS-0, Masa-0) × two conditions (non-irradiated, acutely irradiated) × three Petri dishes per condition × three plants per Petri dish) after adaptation to darkness for at least 30 min. Determination of the minimum levels of actinic light intensity (I_k_) required for subsequent measurements of fluorescence parameters was performed using the method of rapid light curves (RLC) [50,51] and amounted 420 μmol photons m^−2^ × s^−1^. The intensity of saturation pulse was 5250 μmol photons m^−2^ × s^−1^ under duration of 0.8 s. The levels of low frequency measuring light were selected in order to avoid the Kautsky effect and amounted 190 μmol photons m^−2^ × s^−1^.

The following fluorescence parameters were measured in vivo, according to the manufacturer protocol: basic fluorescence yield recorded with low measuring light intensities (F_0_), maximum chlorophyll fluorescence yield (Fm), maximum quantum yield of photochemistry of photosystem II (Fv/Fm), coefficients of photochemical fluorescence (qP and qL), effective photochemical quantum yield of photosystem II (Y(II)), electron transport rate (ETR, μmol photons m^−2^ × s^−1^), coefficient of non-photochemical fluorescence quenching (qN) and non-photochemical fluorescence quenching (NPQ). The photosynthetic parameters were calculated using WinControl v 3.30 software (Heinz Walz GmbH, Effeltrich, Germany). Non-destructive measurements of leaf areas of *A. thaliana* were performed using the open-source software Easy Leaf Area (ELA) [52]. 

### 5.5. Processing of Morphological and Fluorescence Data

Two experiments with independent irradiation of seeds were performed. For each accession, the seeds were sown on four Petri dishes after irradiation. The results presented herein are based on analysis of combined data of the two experiments performed, if not specified otherwise. Statistically significant differences were determined by the Mann–Whitney *U*-test, or Kruskal–Wallis test. Results with *p*-value less or equal to 0.05 were considered significant. Analyses were performed with Statistica 8.0 (Dell Technologies, Round Rock, TX, USA), MS Office Excel 2019 (Microsoft Corporation, Albuquerque, NM, USA) and RStudio v1.4 (R-Tools Technology, Richmond Hill, ON, Canada).

### 5.6. Phytohormone Measurement

The concentrations of the auxins indoleacetic acid (IAA) and indole-3-butyric acid (IBA), zeatin, abscisic acid (ABA), and salicylic acid were assessed in samples of 17-day-old plants using a LC-30 Nexera high-performance liquid chromatograph system (Shimadzu, Kyoto, Japan) as detailed in [53]. The results were processed using LabSolutions software (Shimadzu Corporation, Kyoto, Kyoto Prefecture, Japan). Two to four replicates were used for each experimental point (sample weight of 0.3 g). Each sample was analysed in two technical replicates.

### 5.7. RNA Extraction

Total RNA was extracted from 13-day-old seedlings with a GeneJet Plant RNA Purification Mini Kit (Thermo Fisher Scientific, Waltham, MA, USA) according to the manufacturer protocol. All further sample preparation for Illumina Sequencing was performed by Evrogen company (Moscow, Russian Federation).

### 5.8. Illumina Sequencing

Transcriptome analysis was performed for 12 *A. thaliana* samples: three accessions (Bab-0, VS-0, Masa-0) × two conditions (non-irradiated, acutely irradiated) × two replicates of 100 μg of plant tissue. After the RNA quality control assessment, poly(A) enrichment and synthesis of cDNA with random primers was performed using TruSeq Stranded mRNA kit (Illumina Inc., San Diego, CA, USA). The resulting cDNA was used for preparation of libraries compatible with TruSeq (Illumina Inc., San Diego, CA, USA). The quality of the resulting libraries was checked using Fragment Analyzer Systems (Agilent Technologies, Santa Clara, CA, USA). Quantitative analysis was performed using the qPCR method. After quality control and quantitative assessment, the pool of cDNA libraries was sequenced on Illumina NovaSeq 6000 SP (2 × 150 bp, Illumina Inc., San Diego, CA, USA). As a result, 1.090.535.508 reads were received: 107.927.544 and 190.269.208 reads for acutely irradiated and background plants of Bab-0 accession, respectively; 172.682.154 and 184.025.394 for acutely irradiated and chronically irradiated plants of VS-0 accession, respectively; 178.016.214 and 189.350.394 reads for acutely irradiated and chronically irradiated plants of Masa-0 accession, respectively. 

### 5.9. Data Processing and Functional Analysis

Data pre-processing that included quality control and filtering of sequencing data was performed on Ubuntu 16.04 releases. The quality control checks of raw sequence data were performed using FastQC v 0.11.9 and MultiQC v 1.11. Low-quality reads were filtered with Trimmomatic v 0.39 [54] for paired-end data, using ILLUMINACLIP, HEADCROP, LEADING, TRAILING, SLIDINGWINDOW, and MINLEN parameters. Less than 1% of raw data was found to have a low-quality read and were removed.

The filtered reads were aligned to *A. thaliana* reference genome (TAIR10 from EnsemblPlants database) using HISAT2 v 2.2.1 [55], specifying the strand information (RF) and with –downstream-transcriptome-assembly (--dta) option. StringTie v 2.1.1 was applied to assemble the aligned data into potential transcripts, using as input a HISAT2-processed sample. All transcripts assembly files were merged using merge mode of StringTie for generating a non-redundant set of transcripts. Finally, StringTie was rerun for each HISAT2-processed sample using the -G option with the merged transcripts file, reporting the output tailored for differential expression analysis which was performed with edgeR v 3.34.0 [56]. The three experimental plots were used as a comparison factor, and three comparison groups were formed and the analysis of contrasts of interest was performed. Differential gene expression in *A. thaliana* was presented as the logarithm of fold change (log_2_FC). The accepted level of significance was 0.05 (using the Benjamini and Hochberg correction). Only genes showing log_2_FC > |2| were retained. The transcriptome assembly and differential gene expression analysis were performed using the Galaxy/Europe platform [57]. 

Functional enrichment analysis was performed using Gene Ontology (GO) database with agriGO v 2 web tool. The accepted level of significance was 0.05 (using Fisher as a statistical test method and Yekutielli as an adjustment method).

## Figures and Tables

**Figure 1 plants-11-03142-f001:**
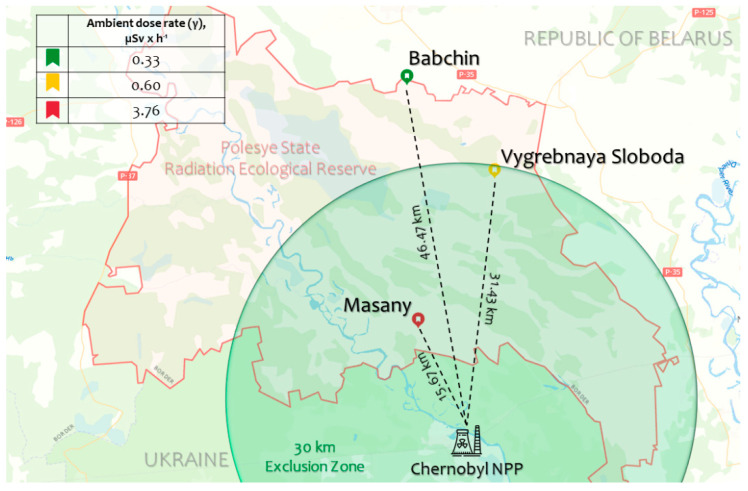
Map of the experimental plots (Babchin, Vygrebnaya Sloboda, and Masany) located in the Polesye State Radiation-Ecological Reserve (Khoiniki, Gomel Region, Republic of Belarus) where *A*. *thaliana* Bab-0, VS-0, and Masa-0 natural accessions, respectively, were collected. The dose rates are represented in Table 1. The map was created using Google Maps (Google LLC, Mountain View, CA, USA), and adapted with Microsoft PowerPoint 2019 (Microsoft Corporation, Albuquerque, NM, USA).

**Figure 2 plants-11-03142-f002:**
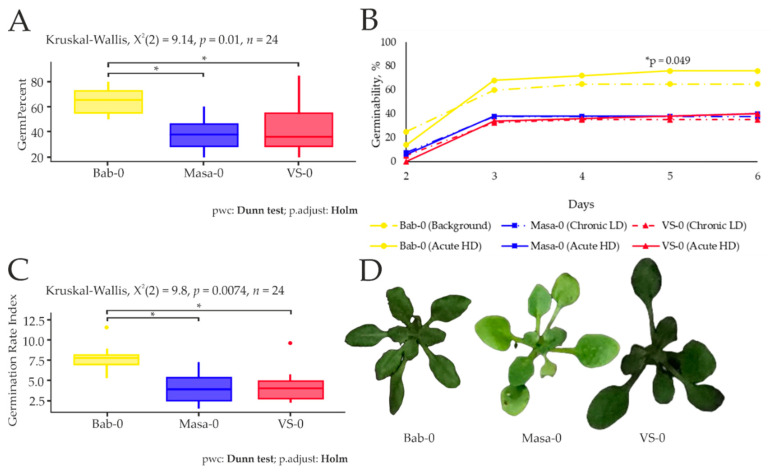
(**A**). Germinability, %, of seeds collected from the Babchin (Bab-0), Masany (Masa-0), Vygrebnaya Sloboda (VS-0) plots. (**B**). Germinability (%, at days 2–6 after seed transfer to the growth chamber. Bab-0 (Background), Masa-0 (Chronic LD) and VS-0 (Chronic LD)—seeds collected from the experimental plots Babchin, Masany, and Vygrebnaya Sloboda, respectively; Bab-0 (Acute HD), Masa-0 (Acute HD) and VS-0 (Acute HD)—seeds collected from the experimental plots and exposed to acute γ-irradiation at a dose of 150 Gy. (**C**). Germination Rate Index, seed × days^−1^, of seeds collected from Babchin (plot with background radiation level) and radioactively contaminated plots Masany and Vygrebnaya Sloboda. (**D**). *A. thaliana* plants from the experimental plots. The graphs were created using RStudio v1.4 (R-Tools Technology, Richmond Hill, ON, Canada). Data are presented for two independent experiments with four biological replicates in each (20 seeds per replicate for each accession). *—significant differences at *p* < 0.05 ((**A**,**C**)—Kruskal–Wallis test with Dunn’s test using the Holm–Bonferroni correction for multiplicity; (**B**)—irradiated compared to control, Mann–Whitney *U*-test).

**Figure 3 plants-11-03142-f003:**
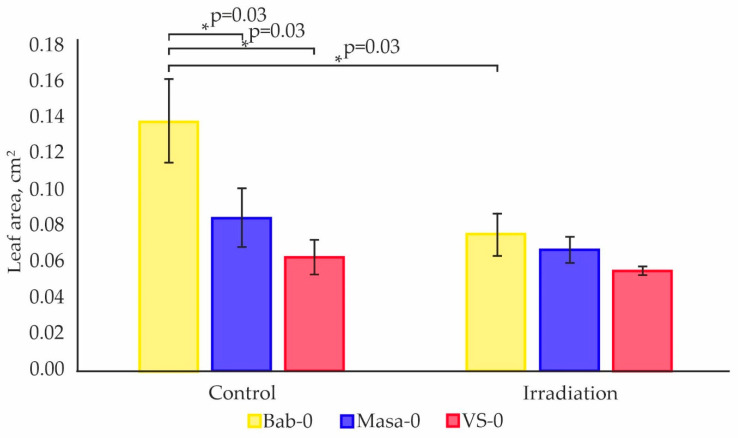
Mean leaf area of one plant (cm^2^) measured on the 11th day of cultivation for four biological replicates (up to 20 plants per replicate). “Control” represents plants from untreated seeds of accessions Bab-0 (Background), Masa-0 (Chronic LD), VS-0 (Chronic LD). “Irradiation“ represents plants grown from seeds exposed to acute γ-irradiation at a dose of 150 Gy (Acute HD). Significant difference with exact *p*-values (Mann–Whitney *U*-test) are denoted by asterisk. The graphs were created using RStudio v1.4 (R-Tools Technology, Richmond Hill, ON, Canada).

**Figure 4 plants-11-03142-f004:**
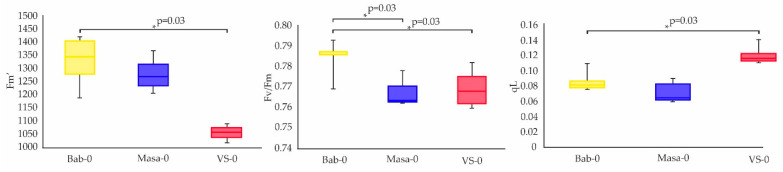
Parameters of photosynthetic activity of 17-day-old plants, grown from untreated seeds of accessions Bab-0 (Background), Masa-0 (Chronic LD), VS-0 (Chronic LD). Fluorescence was measured on the 17th day of cultivation in three plants per replicate for three biological replicates. Significant differences with exact *p*-values (Mann–Whitney *U*-test) are denoted by asterisk. The graphs were created using RStudio v1.4 (R-Tools Technology, Richmond Hill, ON, Canada).

**Figure 5 plants-11-03142-f005:**
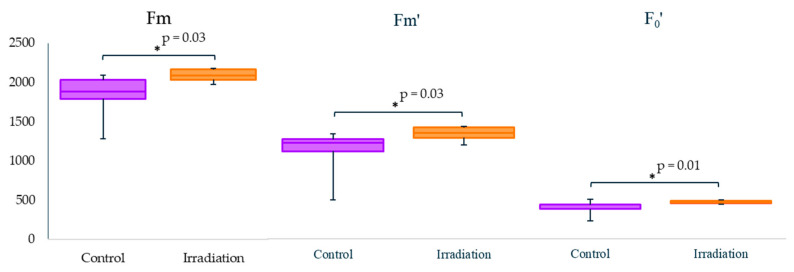
Chlorophyll fluorescence of juvenile Bab-0 plants, measured on the 17th day of cultivation in three plants per replicate for three biological replicates. “Control”—plants grown from untreated Bab-0 (Background) seeds; “Irradiation”—plants grown from seeds exposed to acute γ-radiation at a dose of 150 Gy, Bab-0 (Acute HD). Significant differences with exact *p*-values (Mann–Whitney *U*-test) are denoted by asterisk. The graphs were created using RStudio v1.4 (R-Tools Technology, Richmond Hill, ON, Canada).

**Figure 6 plants-11-03142-f006:**
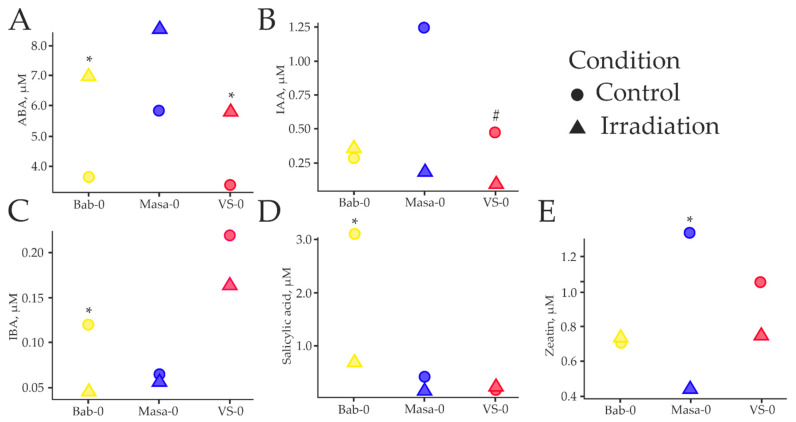
Median values of concentrations of five phytohormones (ABA—abscisic acid (**A**); IAA—indole-3-acetic acid (**B**); IBA—indole-3-butyric acid (**C**); salicylic acid (**D**); zeatin (**E**)) measured on 17th day in plants grown from seeds of Bab-0, VS-0, and Masa-0 accessions. Dots—control plants grown from seeds collected from the experimental plots with background or low-dose chronic irradiation and not exposed to acute γ-irradiation; triangles—plants grown from seeds exposed to acute γ-irradiation at a dose of 150 Gy. Phytohormones were measured in two technical replicates of 2–4 biological replicates composed by a pool of 10 plants each. Significant differences (Mann–Whitney *U*-test) between non-treated and acutely irradiated plants are denoted by * *p* < 0.050; # *p* = 0.052. The graphs were created using RStudio v1.4 (R-Tools Technology, Richmond Hill, ON, Canada).

**Figure 7 plants-11-03142-f007:**
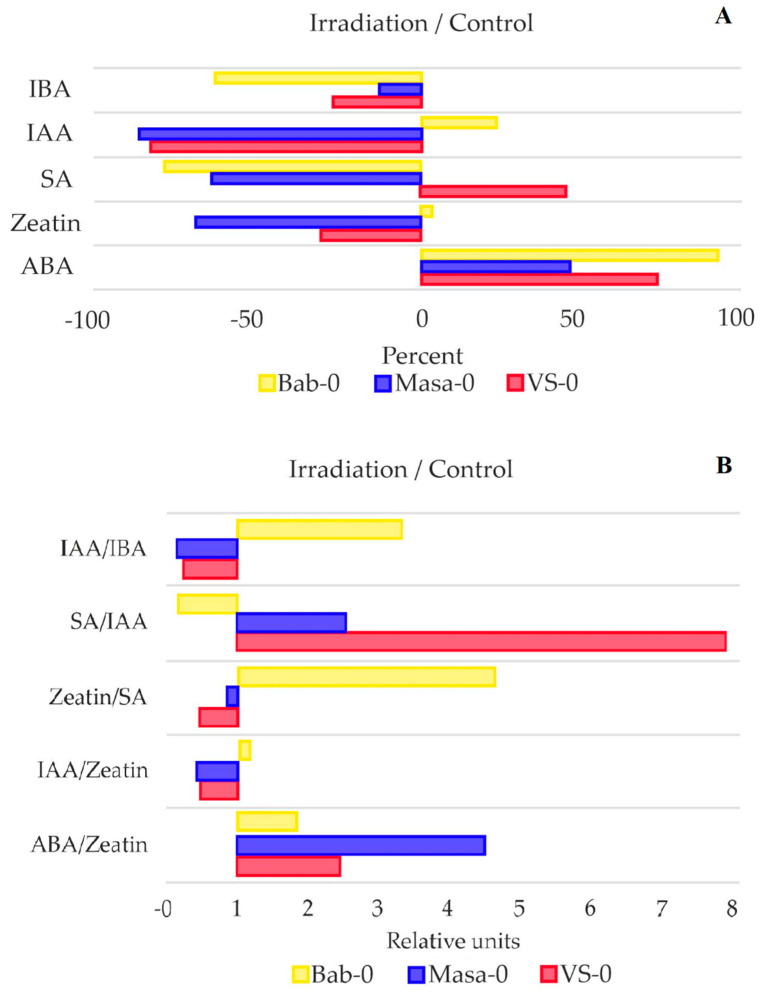
(**A**)—Percentage change in phytohormonal content of acutely irradiated plants to control samples (background Bab-0, chronically irradiated VS-0 and Masa-0). Values <0 denote a decrease in the phytohormone concentration, and >0 show an increase as a result of acute γ-irradiation at a dose of 150 Gy. (**B**)—Phytohormonal ratios of acutely irradiated samples to the same ratio of control samples (background Bab-0 and chronically irradiated VS-0 and Masa-0). Values <1 denote a decrease in the phytohormonal ratio as a result of acute γ-irradiation at a dose of 150 Gy, and values >1 show an increase. IBA—indole-3-butyric acid, IAA—indole-3-acetic acid, SA—salicylic acid, ABA—abscisic acid. The graphs were created using RStudio v1.4 (R-Tools Technology, Richmond Hill, ON, Canada).

**Figure 8 plants-11-03142-f008:**
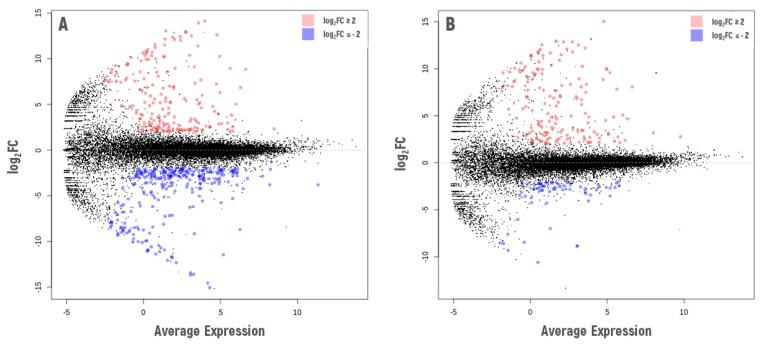
Overview of differential gene expression of 13-day-old *A. thaliana* seedlings recovering from chronic radiation exposure ((**A**). VS-0, and (**B**). Masa-0) in comparison to the background reference accession Bab-0. Log_2_FC—logarithm of gene expression fold change. All plants were grown under control conditions, pooled leaf sample from a Petri dish was used as a biological replicate. Two replicates per condition were used for RNA sequencing. The graphs were created using the edgeR Package for RStudio v1.4 (R-Tools Technology, Richmond Hill, ON, Canada). Graphical changes were made with Microsoft PowerPoint 2019 (Microsoft Corporation, Albuquerque, NM, USA).

**Figure 9 plants-11-03142-f009:**
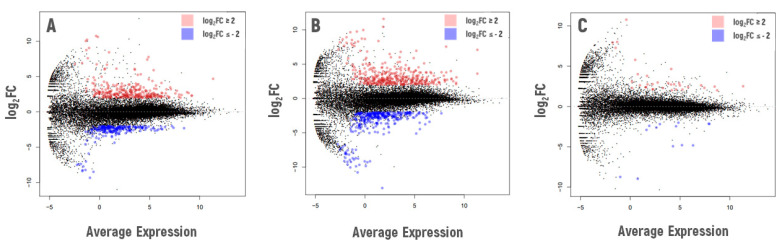
Overview of differential gene expression for additionally irradiated seedlings of *A. thaliana* selected from three experimental plots ((**A**)—Bab-0, (**B**)—VS-0, (**C**)—Masa-0) in comparison with the same accessions without acute irradiation. Log_2_FC—logarithm of gene expression fold change. The graphs were created using the edgeR Package for RStudio v1.4 (R-Tools Technology, Richmond Hill, ON, Canada). Graphical changes were made with Microsoft PowerPoint 2019 (Microsoft Corporation, Albuquerque, NM, USA).

**Figure 10 plants-11-03142-f010:**
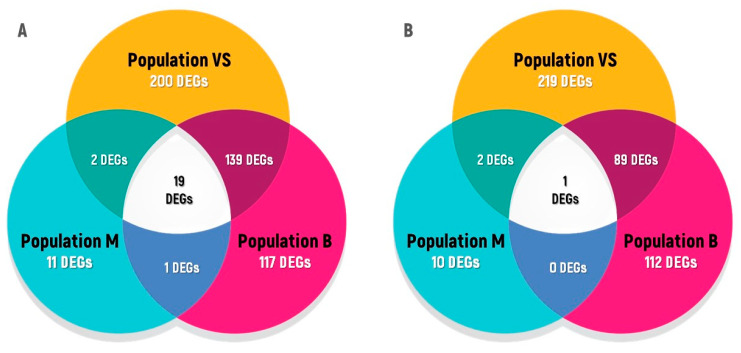
Unique and common responses to acute irradiation of seeds (150 Gy) for seedlings of *A. thaliana* from the CEZ. (**A**)—upregulated genes; (**B**)—downregulated genes. The graph was created using the Microsoft PowerPoint 2019 (Microsoft Corporation, Albuquerque, NM, USA).

**Table 1 plants-11-03142-t001:** Soil properties of the experimental plots.

Experimental Plot (Natural Accession)	pH	HA	TEB	Humus	Av. P_2_O_5_	Av. K_2_O	Av. Ca	Av. Mg	Av. Na
		mg-eqv per 100 g		%	mg/kg		mg-eqv per 100 g		
Babchin (Bab-0)	6.61	6.4	6.1 ± 0.7	1.17 ± 0.01	210.7 ± 5.2	91.0 ± 1.0	3.41 ± 0.20	0.85 ± 0.02	0.61 ± 0.04
Vygrebnaya Sloboda (VS-0)	6.33 ± 0.02	4.3	1.1 ± 0.1	0.60 ± 0.02	47.2 ± 0.6	19.0 ± 0.1	1.06 ± 0.08	0.23 ± 0.01	0.71 ± 0.01
Masany (Masa-0)	5.50 ± 0.01	3.8	1.0 ± 0.4	1.59 ± 0.02	54.5 ± 0.4	16.9 ± 0.2	1.16 ± 0.07	0.35 ± 0.01	0.61

Note: HA—hydrolytic acidity; TEB—total exchangeable bases; Av.—available.

**Table 2 plants-11-03142-t002:** The levels of radioactive contamination at the experimental plots.

Experimental Plot (Natural Accession)	Ambient Dose Rate (γ)	α-Particles Flux Density	β-Particles Flux Density	Activity of ^137^Cs in Soil	Activity of ^90^Sr in Soil
	μSv × h^−1^	min^−1^ × cm^−2^		Bq × kg^−1^	
Babchin (Bab-0)	0.33	1.2	1.2	126 ± 4.5	9.9 ± 1.6
Vygrebnaya Sloboda (VS-0)	0.60	17.0	5.2	1074 ± 31	16.3 ± 2.7
Masany (Masa-0)	3.76	25.3	21.3	11,510 ± 294	233.8 ± 35.4

**Table 3 plants-11-03142-t003:** Set of enriched GO terms in the chronically irradiated *A. thaliana* natural accessions VS-0 and Masa-0 in comparison with the reference accession Bab-0.

**Chronically Irradiated Accession VS-0**		
**GO Terms**	**Up/Downregulated Genes**	***p*-Value**
*Molecular Function*		
protein serine/threonine kinase activity (GO:0004674)	Down	3.9 × 10^−6^
ADP binding (GO:0043531)	Down	3.9 × 10^−6^
ATP binding (GO:0005524)	Down	3.8 × 10^−4^
calcium ion binding (GO:0005509)	Down	3.9 × 10^−2^
*Biological Process*		
protein phosphorylation (GO:0006468)	Down	1.2 × 10^−6^
defence response to bacterium (GO:0042742)	Down	5.6 × 10^−3^
systemic acquired resistance (GO:0009627)	Down	3.0 × 10^−3^
signal transduction (GO:0007165)	Down	2.6 × 10^−3^
response to ozone (GO:0010193)	Down	2.6 × 10^−3^
response to chitin (GO:0010200)	Down	1.0 × 10^−2^
positive regulation of innate immune response (GO:0045089)	Down	1.0 × 10^−2^
response to UV-B (GO:0010224)	Down	1.7 × 10^−2^
**Chronically Irradiated Accession Masa-0**		
**GO Terms**	**Up/Downregulated Genes**	***p*-Value**
*Molecular Function*		
ADP binding (GO:0043531)	Up	2.3 × 10^−4^
protein serine/threonine kinase activity (GO:0004674)	Down	4.8 × 10^−2^
carbohydrate binding (GO:0030246)	Down	4.8 × 10^−2^

**Table 4 plants-11-03142-t004:** The selected enriched GO terms for the seedlings of acutely irradiated seeds in comparison with seeds of the same accession without additional irradiation.

**Acutely Irradiated Plants of the Accession Bab-0**		
**GO Terms**	**Up/Downregulated Genes**	***p*-Value**
*Molecular Function*		
glutathione transferase activity (GO:000258)	Up	2.6 × 10^−4^
kinase activity (GO:0016301)	Up	3.5 × 10^−4^
xyloglucan: xyloglucosyl transferase activity (GO:0016762)	Up	2.3 × 10^−3^
oxidoreductase activity, act. on a sulphur group of donors (GO:0016667)	Up	4.3 × 10^−2^
hydrolase activity, hydrolysing O-glycosyl compounds (GO:0046527)	Up/Down	2.4 × 10^−2^/2.9 × 10^−2^
peroxidase activity (GO:0004601)	Down	9.9 × 10^−5^
heme biding (GO:0020037)	Down	2.1 × 10^−4^
copper ion binding (GO:0005507)	Down	2.8 × 10^−3^
oxidoreductase activity, oxidizing metal ions (GO:0016722)	Down	2.4 × 10^−3^
*Biological Process*		
response to toxic substance (GO:0009636)	Up	9.5 × 10^−6^
response to salicylic acid (GO:0009751)	Up	8.3 × 10^−5^
toxic catabolic process (GO:0009407)	Up	4.4 × 10^−5^
defence response to bacterium (GO:0042742)	Up	4.0 × 10^−5^
response to wounding (GO:0009611)	Up	2.5 × 10^−5^
response to jasmonic acid (GO:0009753)	Up	1.5 × 10^−4^
xyloglucan metabolic process (GO:0010411)	Up	8.8 × 10^−3^
response to virus (GO:0009615)	Up	6.3 × 10^−3^
response to abscisic acid (GO:0009737)	Up	6.0 × 10^−3^
defence response to fungus (GO:0050832)	Up	1.6 × 10^−3^
oxidation-reduction process (GO:0055114)	Up/Down	2.7 × 10^−2^/1.9 × 10^−6^
leaf senescence (GO:0010150)	Up	2.4 × 10^−2^
response to salt stress (GO:0009651)	Up	1.4 × 10^−2^
response to water deprivation (GO:0009414)	Up	1.7 × 10^−2^
suberin biosynthetic process (GO:0010345)	Down	1.4 × 10^−6^
hydrogen peroxide catabolic process (GO:0042744)	Down	4.5 × 10^−5^
response to oxidative stress (GO:0006979)	Down	5.7 × 10^−4^
root morphogenesis (GO:0010015)	Down	2.7 × 10^−4^
lignin metabolic process (GO:0009808)	Down	5.2 × 10^−3^
plant-type cell wall organization (GO:0009664)	Down	4.1 × 10^−3^
*Cellular Component*		
plasma membrane (GO:0005886)	Up	8.0 × 10^−4^
apoplast (GO:0048046)	Up	2.7 × 10^−3^
integral component of membrane (GO:0016021)	Up	1.3 × 10^−3^
cell wall (GO:0005618)	Up	1.1 × 10^−2^
extracellular region (GO:0005576)	Down	5.2 × 10^−5^
**Acutely Irradiated Plants of the Accession VS-0**		
**GO Terms**	**Up/Downregulated Genes**	***p*-Value**
*Molecular Function*		
xyloglucan: xyloglucosyl transferase activity (GO:0016762)	Up	2.2 × 10^−3^
peroxidase activity (GO:0004601)	Down	3.6 × 10^−8^
heme binding (GO:0020037)	Down	3.8 × 10^−6^
quercetin 3-O-glucosyltransferase activity (GO:0080043)	Down	3.8 × 10^−2^
quercetin 7-O-glucosyltransferase activity (GO:0080043)	Down	3.8 × 10^−2^
symporter activity (GO:0015293)	Down	2.7 × 10^−2^
*Biological Process*		
response to jasmonic acid (GO:0009753)	Up	3.9 × 10^−6^
defence response to fungus (GO:0050832)	Up	3.3 × 10^−6^
response to salicylic acid (GO:0009751)	Up	1.7 × 10^−6^
defence response to bacterium (GO:0042742)	Up	1.4 × 10^−5^
response to reactive oxygen species (GO:0000302)	Up	6.3 × 10^−4^
response to salt stress (GO:0009651)	Up	5.2 × 10^−3^
xyloglucan metabolic process (GO:0010411)	Up/Down	4.8 × 10^−3^/3.0 × 10^−2^
cellular calcium ion homeostasis (GO:0006874)	Up	3.0 × 10^−3^
lignin metabolic process (GO:0009808)	Up/Down	3.8 × 10^−2^/1.1 × 10^−2^
response to abscisic acid (GO:0009737)	Up	1.3 × 10^−2^
toxin catabolic process (GO:0009407)	Up	1.2 × 10^−2^
hydrogen peroxide metabolic process (GO:0042744)	Down	7.6 × 10^−7^
oxidation-reduction process (GO:0055114)	Down	5.8 × 10^−5^
response to hypoxia (GO:0001666)	Down	1.9 × 10^−4^
response to oxidative stress (GO:000513)	Down	5.1 × 10^−3^
plant type cell wall organization (GO:0009664)	Down	2.2 × 10^−3^
root morphogenesis (GO:0010015)	Down	3.0 × 10^−2^
phenylpropanoid biosynthetic process (GO:0009699)	Down	3.0 × 10^−2^
cellular response to starvation (GO:0009267)	Down	2.6 × 10^−2^
*Cellular Component*		
cell wall (GO:0005618)	Up/Down	3.5 × 10^−6^/7.6 × 10^−6^
plasma membrane (GO:0005886)	Up	4.5 × 10^−3^
extracellular space (GO:0005615)	Down	3.3 × 10^−2^
**Acutely Irradiated Plants of the Accession Masa-0**		
**GO Terms**	**Up/Downregulated Genes**	***p*-Value**
*Biological Process*		
innate immune response (GO:0045087)	Up	2.7 × 10^−3^
regulation of growth (GO:0040008)	Up	1.7 × 10^−3^
hormone-mediated signaling pathway (GO:0009755)	Up	1.2 × 10^−2^
defence response to another organism (GO:0098542)	Up	1.4 × 10^−2^
*Cellular Component*		
cell wall	Up	3.5 × 10^−2^

**Table 5 plants-11-03142-t005:** Unique DEGs in accession Masa-0 from the most contaminated plot in response to acute and chronic irradiation.

Gene	Description	Acute Irradiation ^1^	Chronic Irradiation ^2^
		log_2_FC	
*AT4G05235*	Long non-coding RNA	10.77	−8.45
*AT1G15640*	Transmembrane protein	7.93	−8.59
*AT5G14160*	F-box family protein	7.28	−7.58
*AT3G21330*	Basic helix–loop–helix (bHLH) DNA-binding superfamily protein	4.00	
*AT5G18010*	SAUR-like auxin-responsive protein family	3.10	
*AT5G39860*	Basic helix–loop–helix (bHLH) DNA-binding family protein	2.74	
*AT5G18050*	SAUR-like auxin-responsive protein family	2.63	
*AT1G52400*	Beta glucosidase 18	2.47	
*AT5G18080*	SAUR-like auxin-responsive protein family	2.39	
*AT5G18030*	SAUR-like auxin-responsive protein family	2.39	
*AT3G09960*	Calcineurin-like metallo-phosphoesterase superfamily protein	2.31	−2.80
*AT3G57520*	Seed imbibition 2	−2.14	
*AT3G30775*	Proline oxidase	−2.17	
*AT5G41080*	PLC-like phosphodiesterases superfamily protein	−2.45	
*AT1G02820*	Late embryogenesis abundant 3 (LEA3) family protein	−2.61	
*AT4G04223*	Other RNA	−2.90	4.90
*AT3G60950*	C2 calcium/lipid-binding endonuclease/exonuclease/phosphatase	−4.80	−2.67
*AT3G62460*	Putative endonuclease or glycosyl hydrolase	−4.82	4.68
*AT3G61030*	Calcium-dependent lipid-binding (CaLB domain) family protein	−4.93	−2.88
*AT2G29000*	Leucine-rich repeat protein kinase family protein	−8.74	8.74
*AT2G27402*	Hypothetical protein	−8.97	12.56

^1^ Acute irradiation—comparison of transcriptomes of 150 Gy-irradiated Masa-0 plants vs. Masa-0 plants without additional irradiation. ^2^ Chronic irradiation—comparison of transcriptomes of Masa-0 plants without additional irradiation vs. Bab-0 background plants without additional irradiation.

## Data Availability

The processed sequencing files are available at the Sequence Read Archive (BioProject PRJNA787059).

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
