# Peer review of "Arabidopsis thaliana Accessions from the Chernobyl Exclusion Zone Show Decreased Sensitivity to Additional Acute Irradiation"

_plants, 2022, doi:10.3390/plants11223142_

Round 1

Reviewer 1 Report

The manuscript “Arabidopsis thaliana accessions from the Chernobyl exclusion 2 zone show decreased sensitivity to additional acute irradiation” contains many interesting results. But the text is sometimes unclear and needs some correction.

Results:

The manuscript contains many results but percentage values are missing in the whole part.

Fig. 6B Why is there # and not *?

Discussion:

The discussion is poor and mainly repeated the results. The authors did not explain their obtained results.

Conclusion

The conclusion again contains a summary of the results without an explanation of them in the defence mechanisms against radiation.

Author Response

The authors of the manuscript would like to thank the three Reviewers for thoroughly checking the manuscript and for the valuable comments and suggestions, we greatly appreciated them. The design of the experiment is complex, and your remarks really helped us to improve the presentation and discussion of our data. We took into account all the points raised and provided point-by-point response to each of them. All changes made in the manuscript are highlighted in a version “manuscript_revised_changes highlighted”.

Response to the Reviewer 1

The manuscript “Arabidopsis thaliana accessions from the Chernobyl exclusion 2 zone show decreased sensitivity to additional acute irradiation” contains many interesting results. But the text is sometimes unclear and needs some correction.

We appreciate the remark, and we took it into account in the revised version.

The manuscript contains many results but percentage values are missing in the whole part.

We are sorry, but we did not completely understand this comment. The percentage values are provided for the phytohormonal part only.

Fig. 6B Why is there # and not *?

The sign was used to show the tendency. For clarification, a description was added to the legend of Fig. 6B - # p=0.052.

The discussion is poor and mainly repeated the results. The authors did not explain their obtained results.

Thank you for your comment. We agree that the discussion had room for improvement, which was explored in this latest version.

The conclusion again contains a summary of the results without an explanation of them in the defence mechanisms against radiation.

Thank you, we rewrote the conclusion.

Reviewer 2 Report

This is a well-written paper on an interesting topic, comparing several measures in arabidopsis collected at different levels of nuclear contamination. However, there is one very significant issue in the experimental set up that make it difficult to recommend for publication. Perhaps they can clear that up. There are also a few more minor issues to clarify, further below. 

The major issue is sampling. They selected just three cites for analysis at different distances from the source, and collected 'ascensions' from these cites. They show measurements of radioactivity in the figure, and of soil properties and heavy metals in the supplemental information, Tables S1.

Contrary to their statement on line 75 "but similar soil composition and weather conditions", their data show significantly different soil properties among the cites, including substantially higher pH, base cations, P, K, Ca and Mg in the Babchin soils than in the other soils that are more contaminated with radiation. This makes it very difficult to say that all of their measured differences are due to radiation exposure or to other variables.

Similarly, numerous other variables were evidently not measured or at least not presented here. For example, they do not report soil type (e.g. texture) and cite slope and aspect, and relative shade vs. sun exposure. All of these would combine in a drought-prone vs. less drought-prone variables. And reporting these would go a long way in better getting a sense of whether and to what extent their measured differences were due to radiation or due to other abiotic factors that differentiate these cites.   In fact, several of their measurements (e.g. Fv/Fm, ABA and other phytohormones, antioxidant genes, etc...) have been previously shown to elevated in response to drought, salt, temperature stress, etc. Could we just be seeing that here as well, with poorer soils with more likelihood of drought, closer to the power plant?

The authors should carefully address these issues in the paper, and not hide their soil comparisons supplemental info. The comparison may still be valid, but they'll need to show the reader that it is, rather than skimming over this issue entirely.

More minor issues that can be more easily addressed in a re-write:

1. Figs 4 and 5. In figure 4 it is unclear why these particular fluorescent measures are shown or what they mean. We know the meaning of Fv/Fm, but the authors don't explain what changes in these other measures after acute irradiation might mean. Similarly, why only show the data for the least contaminated cite? (Isn't it part of the point to show that the plants from the more contaminated cites do not respond as much to acute radiation exposure? you can show that here too?) Fig. 5. It is unclear from reading the paper and the caption, and comparing that to what is reported in Table S2, whether the data is from acute-radiation exposed plants, or not (their analysis in Table S2 is about acute exposure?) 

2. Fig. 6. It is unclear what differences are significant. Are they saying there's a difference in background phytohormone concentrations between plants from the different cites, for that it's different between untreated and irradiated plants of the same type? I also would prefer a bar chart with error bars, or box and whisker plots (as in Fig. 5), since it's difficult to gauge the variability of these measurements.  Fig 7 and 8 are interesting, though redundant. Also, without error bars and/or some indication of significances, it's difficult to draw much from these graphs.

3. I really like the presentation in figures 9 and 10! (and would probably be as happy if the prior figures were removed all together and just these data centered in the paper). It would be useful if a sentence or two were added to the caption explaining the axes - what is "log2FC"? (I know that you know, but it's easy to add a few words of explanation for us old school physiologists!) 

The tables discussing the GO terms and major up and down items seem well-done. 

Author Response

The authors of the manuscript would like to thank the three Reviewers for thoroughly checking the manuscript and for the valuable comments and suggestions, we greatly appreciated them. The design of the experiment is complex, and your remarks really helped us to improve the presentation and discussion of our data. We took into account all the points raised and provided point-by-point response to each of them. All changes made in the manuscript are highlighted in a version “manuscript_revised_changes highlighted”.

Response to the Reviewer 2

This is a well-written paper on an interesting topic, comparing several measures in arabidopsis collected at different levels of nuclear contamination. However, there is one very significant issue in the experimental set up that make it difficult to recommend for publication. Perhaps they can clear that up. There are also a few more minor issues to clarify, further below. 

Thank you for directing us to some interesting interpretation – we hope that we managed to address all the points.

The major issue is sampling. They selected just three cites for analysis at different distances from the source, and collected 'ascensions' from these cites.

Due to natural availability, the accessions were only found in three plots within the accessible areas in the Polesye Radiation-Ecological Reserve forest, which comprises a significant part of the Chernobyl exclusion zone. Arabidopsis prefers anthropogenic-disturbed areas, reason why this pecies is difficult to find in forest reserves. Also, this observation supports  that these populations are isolated from each other, meaning that genetic drift is not the main factor contributing to the observed responses.

Contrary to their statement on line 75 "but similar soil composition and weather conditions", their data show significantly different soil properties among the cites, including substantially higher pH, base cations, P, K, Ca and Mg in the Babchin soils than in the other soils that are more contaminated with radiation. This makes it very difficult to say that all of their measured differences are due to radiation exposure or to other variables.

Although there are indeed differences which cannot be controlled in experiments with natural populations, the pH of soils is still optimal for plants (soils can’t be attributed to acidic or alkaline), and the nutrition factor is poor, what is typical for soils of the region. Among the nutrients, the most relevant for the present study would be potassium because it competes with caesium on root uptake. Nevertheless, our transcriptional data shows that the radiation effect is traceable, thus representing the major component for the observed differences. These ideas have been added to the manuscript, and the discussion has been updated accordingly.

Move soil analysis to the Table inside the manuscript.

We added soil analysis as Table 1.

Similarly, numerous other variables were evidently not measured or at least not presented here. For example, they do not report soil type (e.g. texture) and cite slope and aspect, and relative shade vs. sun exposure. All of these would combine in a drought-prone vs. less drought-prone variables. And reporting these would go a long way in better getting a sense of whether and to what extent their measured differences were due to radiation or due to other abiotic factors that differentiate these cites. 

Indeed, when working with field samples, we cannot completely exclude some confounding effects on the response profiles, reason why we try to minimize them. We adopt a series of controls for improving the support of the data: 1. the control seeds were also collected from the field, from the closest area with background IR levels; 2. the experimental plots were visibly similar (open areas, uniform insolation); 3. the seeds were collected under the same weather conditions, which was also stable during all the week preceding the sampling; 4 the samples from different plots were collected within two days, which means they also have approximately the same age when considering the short life-cycle of A. thaliana; 5. for the transcriptome analysis, all plants were germinated and grown under the same controlled conditions; 6. for a better support of IR-dependent responses, we focus on patterns that overlap between organisms coming from contaminated areas. Furthermore, if the soil was the main driver of the differences, it would be expected that VS-0 (low contamination) and Masa-0 (higher contamination) would should a more similar transcriptome profile after acute high-dose irradiation. However, VS-0's profile was closer to the control Bab-0. We added the soils type and information on slope in the results (highlighted). We did not measure PAR of the plots. The collected plants were growing on similar plots, in open areas with uniform insolation during the daytime. We added this information to the Materials and Methods section.

In fact, several of their measurements (e.g. Fv/Fm, ABA and other phytohormones, antioxidant genes, etc...) have been previously shown to elevated in response to drought, salt, temperature stress, etc. Could we just be seeing that here as well, with poorer soils with more likelihood of drought, closer to the power plant?

The overlap of the measured parameters between ionizing radiation and other climate-related abiotic stresses is indeed relevant, and it has been observed for other plant species on different plots (please refer to Duarte et al., 2019 and Volkova et al., 2021), but also during in vitro experiments under controlled conditions (Volkova et al., 2019, and Volkova et al., 2020). Furthermore, it is also important to highlight that all plants used in this work were grown from seeds under controlled laboratory conditions, which helps us to exclude the possible interference of weather difference in situ for these adult plants.

Figs 4 and 5. In figure 4 it is unclear why these particular fluorescent measures are shown or what they mean. We know the meaning of Fv/Fm, but the authors don't explain what changes in these other measures after acute irradiation might mean.

Thank you, we added this information to the results.

Similarly, why only show the data for the least contaminated cite? (Isn't it part of the point to show that the plants from the more contaminated cites do not respond as much to acute radiation exposure? you can show that here too?)

We introduced a clearer link to the table S2, which is related to the measurements taken on plants from VS-0 and Masa-0 accessions. Due to the number of measured parameters, we had to opt for showing only part of them in the main text.

Fig. 5. It is unclear from reading the paper and the caption, and comparing that to what is reported in Table S2, whether the data is from acute-radiation exposed plants, or not (their analysis in Table S2 is about acute exposure?) 

We corrected Table S2 and now it is more straightforward. It contains data on both acute and chronic irradiation.

Fig. 6. It is unclear what differences are significant. Are they saying there's a difference in background phytohormone concentrations between plants from the different cites, for that it's different between untreated and irradiated plants of the same type?

Thank you, we indeed did not clarify that. It has been corrected.

I also would prefer a bar chart with error bars, or box and whisker plots (as in Fig. 5), since it's difficult to gauge the variability of these measurements. 

We made a whisker plot, but following your concern we think it did not reflect what we wanted to show – the changes of median values between chronically and acutely irradiated plants. Values on whisker plot overlaps making it hard to read, so we would like to ask to keep this figure as it is now.

Fig 7 and 8 are interesting, though redundant. Also, without error bars and/or some indication of significances, it's difficult to draw much from these graphs.

Since it’s comparison of medians, there are no error bars. To diminish redundancy, we combined these Figures to the one Figure 7.

  1. I really like the presentation in figures 9 and 10! (and would probably be as happy if the prior figures were removed all together and just these data centered in the paper).

Thank you! We tried to diminish the number of prior Figures to several panels, but the way of presentation in the Plant journal makes it really bad for reading.

It would be useful if a sentence or two were added to the caption explaining the axes - what is "log2FC"? (I know that you know, but it's easy to add a few words of explanation for us old school physiologists!) 

Sorry that we missed it – the caption is added.

Reviewer 3 Report

This study deals with the effect of radiation on the Arabidopsis plant. The manuscript is well-detailed and the research is well-described with appropriate statistical analysis included. I have two main remarks.

The introduction is quite short and contains several references on own previous studies. Although the Chernobyl area is quite unique, there is a need to refer also to international literature. Further to that the novelty of the research should be described and emphasized better.

The other remark is related to the discussion part. It feels like a short summary of the results and the comparison with international references is mostly missing. I assume that the uniqueness of the research makes this part harder to accomplish. Comparisons could be made with other studies on the effect of radioactivity on plants. Novel findings should be emphasized more. In the conclusion, part novel findings could be also underlined and explained better. The conclusion part could be made more compact.

Author Response

The authors of the manuscript would like to thank the three Reviewers for thoroughly checking the manuscript and for the valuable comments and suggestions, we greatly appreciated them. The design of the experiment is complex, and your remarks really helped us to improve the presentation and discussion of our data. We took into account all the points raised and provided point-by-point response to each of them. All changes made in the manuscript are highlighted in a version “manuscript_revised_changes highlighted”.

Response to the Reviewer 3

This study deals with the effect of radiation on the Arabidopsis plant. The manuscript is well-detailed and the research is well-described with appropriate statistical analysis included. I have two main remarks.

Thank you for your positive assessment of our manuscript! We took into account all the point raised.

The introduction is quite short and contains several references on own previous studies. Although the Chernobyl area is quite unique, there is a need to refer also to international literature. Further to that the novelty of the research should be described and emphasized better.

We paid more attention to the novelty of the work. In introduction, we mentioned in all the works we found regarding Arabidopsis studies in Chernobyl, from different scientific groups, including international. If we really missed something, could you please suggest the doi of the relevant articles? We will add them to the introduction.

The other remark is related to the discussion part. It feels like a short summary of the results and the comparison with international references is mostly missing. I assume that the uniqueness of the research makes this part harder to accomplish. Comparisons could be made with other studies on the effect of radioactivity on plants. Novel findings should be emphasized more.

Thank you, we worked more on the Discussion and substantially improved it.

In the conclusion, part novel findings could be also underlined and explained better. The conclusion part could be made more compact.

The conclusion was almost completely redone.

Round 2

Reviewer 1 Report

The text in the part Results contain only the terms “decrease” and “increase” without information about how many parameter decreases/increases. It is not clear from the graph sometimes. Therefore, I suggest inserting percentage values into the text. (Example: Germination rate Index was lower in VS-0 about XY % and Masa-0 about XY% compared to the Bab-0.)

Author Response

Dear Reviewer,

Thank you very much, we improved the presentation of the results accordingly (please see subsections 2.2-2.5)

Reviewer 2 Report

The changes made, and the arguments made in the rebuttal letter, all make sense. This is a nice piece of research, that I find ready for publication in Plants. 

Author Response

Dear Reviewer,

Thank you for the comments that helped us substantially improve our manuscript.

Reviewer 3 Report

I accept the changes and the answers. No further comments.

Author Response

(The authors gave the same response as above.)
